Would future climate warming cause zoonotic diseases to spread over long distances?

Bu Fan 1 2 3
Yue Xiuxian 4
Sun Shanshan 1 2 3
Jin Yongling 1 2 3
Li Linlin 1 2 3
Li Xin 1 2 3
Zhang Rong 1 2 3
Shang Zhenghaoni 1 2 3
Yan Haiwen 1 2 3
Zhang Haoting 1 2 3
Yuan Shuai yuanshuai2020@163.com 1 2 3
Wu Xiaodong 1 2 3
Fu Heping fuheping@126.com 1 2 3
1 Key Laboratory of Grassland Resources, Ministry of Education , Hohhot , China
2 Key Laboratory of Grassland Rodent Ecology and Pest Controlled, Inner Mongolia , Hohhot , China
3 College of Grassland Resources and Environment, Inner Mongolia Agricultural University , Hohhot , China
4 Inner Mongolia Autonomous Region Forestry and Grassland Monitoring Planning Institute , Hohhot , China
Sunny Armando
Electronic publication date: 2024 Feb 21
Publication date: 2024
Volume: 12
Electronic Location ID: e16811
Received 2023 Apr 5; Accepted 2023 Dec 29
Copyright: ©2024 Bu et al.
Copyright year: 2024
Copyright holder: Bu et al.
License: This is an open access article distributed under the terms of the Creative Commons Attribution License, which permits unrestricted use, distribution, reproduction and adaptation in any medium and for any purpose provided that it is properly attributed. For attribution, the original author(s), title, publication source (PeerJ) and either DOI or URL of the article must be cited.
License URL: https://creativecommons.org/licenses/by/4.0/

Keywords: Dipus sagitta, Potential distribution area, Climate change, Hazard dispersal

Funding: Basic scientific research business expenses of universities directly under Inner Mongolia Autonomous Region BR22-13-07 BR220106 BR221037 Chinese Academy of Sciences 32060395 32060256 32090024 Major Science and Technology Project of Inner Mongolia Autonomous Region 2021ZD0006 Pest Diversity Survey in the Western section of Agro-Pastoral Ecotone in North China 2019FY100304 Natural Science Foundation of Inner Mongolia 2019MS03012 Science and Technology Project of Inner Mongolia Autonomous Region 2021GG0108 Program for Young Talents of Science and Technology in Universities of Inner Mongolia Autonomous Region NJYT22044 Grassland Ecological protection and Restoration Treatment Subsidy, Inner Mongolia Autonomous Region Postgraduate Research Innovation Funding Project B20231089Z This study was supported by the Basic scientific research business expenses of universities directly under Inner Mongolia Autonomous Region (BR22-13-07, BR220106,BR221037), Chinese Academy of Sciences (32060395,32060256, 32090024), the Major Science and Technology Project of Inner Mongolia Autonomous Region (2021ZD0006), the Pest Diversity Survey in the Western section of Agro-Pastoral Ecotone in North China (2019FY100304), Natural Science Foundation of Inner Mongolia (2019MS03012), the Science and Technology Project of Inner Mongolia Autonomous Region (2021GG0108), the Program for Young Talents of Science and Technology in Universities of Inner Mongolia Autonomous Region (NJYT22044), and the Grassland Ecological protection and Restoration Treatment Subsidy, Inner Mongolia Autonomous Region Postgraduate Research Innovation Funding Project (B20231089Z). The funders had no role in study design, data collection and analysis, decision to publish, or preparation of the manuscript.

==============================
Dipus sagitta is a major rodent found in arid environments and desert areas. They feed on plant seeds, young branches and some small insects, and have hibernating habits. Peak Dipus sagitta numbers impact the construction of the plant community in the environment, but also have a human impact as these rodents carry a variety of parasitic fleas capable of spreading serious diseases to humans. Based on 216 present distribution records of Dipus sagitta and seven environmental variables, this article simulates the potential distribution of Dipus sagitta during the Last Glacial Maximum, the mid-Holocene, the present and the future (2070s, RCP4.5, RCP8.5). This study also analyzes the geographic changes of the population distribution and evaluates the importance of climate factors by integrating contribution rate, replacement importance value and the jackknife test using the MaxEnt model. In this study, we opted to assess the predictive capabilities of our model using the receiver operating characteristic (ROC) and partial receiver operating characteristic (pROC) metrics. The findings indicate that the AUC value exceeds 0.9 and the AUC ratio is greater than 1, indicating superior predictive performance by the model. The results showed that the main climatic factors affecting the distribution of the three-toed jerboa were precipitation in the coldest quarter, temperature seasonality (standard deviation), and mean annual temperature. Under the two warming scenarios of the mid-Holocene and the future, there were differences in the changes in the distribution area of the three-toed jerboa. During the mid-Holocene, the suitable distribution area of the three-toed jerboa expanded, with a 93.91% increase in the rate of change compared to the Last Glacial Maximum. The size of the three-toed jerboa’s habitat decreases under both future climate scenarios. Compared to the current period, under the RCP4.5 emission scenario, the change rate is −2.96%, and under the RCP8.5 emission scenario, the change rate is −7.41%. This indicates a trend of contraction in the south and expansion in the north. It is important to assess changes in the geographic population of Dipus sagitta due to climate change to formulate population control strategies of these harmful rodents and to prevent and control the long-distance transmission of zoonotic diseases.

Introduction

Climate is considered to be the most important environmental factor in determining species distribution. Climate change has a huge impact on species distribution and biodiversity, and changes in species distribution patterns can also reflect historical climate change trends (Descombes et al., 2015; Allen & Lendemer, 2016). The Fifth Assessment Report of the Intergovernmental Panel on Climate Change (IPCC) pointed out that the average temperature of the Earth’s surface continues to rise as greenhouse gas emissions increase (Fahrig, 2003; Allen et al., 2014). As the global climate warms, suitable habitats for many species are reduced and lost, leading to a dramatic decline in the Earth’s biodiversity (O’Connor, Bojinski & Rsli, 2019; Barnosky, 2006). The reduction and extinction of beneficial animals and protected animal species is a serious, ongoing problem (Huang & Wang, 2019). Another serious problem triggered by climate change is the potential increase in new animal-to-animal and animal-to-human disease transmission caused by the diffuse movement of pest distribution areas, leading to new ecological and human health risks (Carlson et al., 2022). Understanding the impact of climate change on species distribution patterns will help elucidate reasons for habitat changes in the evolutionary history of species. This knowledge will provide an important basis for analyzing and predicting changing species distribution patterns under global warming conditions and inform the development of control strategies for harmful species.

Species distribution models (SDMs), such as the MaxEnt model (maximum entropy model), have been widely used to study the impact of climate change on the potential geographic distribution of species (Romain, Vincent & Jean-Claude, 2012; Su et al., 2015). The MaxEnt model can calculate the maximum entropy of species distribution probability using incomplete species distribution data and environmental data, enabling the prediction of the potential distribution range of the species (Phillips, Anderson & Schapire, 2006) . Previous studies have shown that the MaxEnt model is suitable for species distribution modeling (Peterson, Pape & Eaton, 2007; Li et al., 2009; Robert et al., 2005; Barbosa & Schneck, 2015; Elith, Phillips & Hastie, 2015). Using MaxEnt predictions and ArcGIS software, researchers can identify distribution points that have higher ecological species stability under conditions of global warming and infer the suitable habitat range and change trend of the species (Chan, Brown & Yoder, 2011).

There are many published studies on changes in the suitable distribution areas of different species under different emission scenarios, which have been used to predict: the response of species to climate change, the expansion trends of alien invasive species, species interaction, and impacts on genetic diversity (Urbani, D’Alessandro & Biondi, 2017; Zhang et al., 2021; Pauls et al., 2013). These studies provide the empirical basis for an in-depth understanding of climate change and its impact on pest expansion. Studies have shown that climate change can change the suitable distribution area of species. For example, Wang et al. (2022) predicted the potential distribution area of Meriones meridianus under different future scenarios and found that an increase in temperature and precipitation could change the distribution of food resources and the degree of competition among different populations, leading to a significant reduction in the suitable distribution area of Meriones meridianus. Another future species distribution prediction study found that the overall distribution area of Allactaga firouzi will not change significantly, but the suitable distribution area in China, Kumanstan, and other countries will become smaller and the suitable distribution area in Mongolia, Kazakhstan, and other countries will expand (Mohammadi et al., 2019). The suitable distribution area of Rhombomys opimus will both gradually decrease in overall area and shift to higher latitudes (Wen et al., 2022). Climate change is predicted to change species habitats and ecological processes, and promote species migration (Hill, Griffiths & Thomas, 2011). Bai et al. (2022) found a correlation between temperature factors and local disappearance and migration of Lasiopodomys brandtii. With temperature increases of about 0.36 °C every 10 years, the southern boundary of the Lasiopodomys brandtii distribution area moved at least 495 km to the northern high latitude area. These studies all show that climate change impacts the suitable distribution area of species as these species are forced to adjust their own distribution patterns based on environmental and ecological changes (Schloss, Nuñez & Lawler, 2012; Walther et al., 2002; Parmesan & Yohe, 2003; Lioubimtseva & Henebry, 2009; McLaughlin et al., 2002).

The three-toed jerboa (Dipus sagitta) is a burrowing rodent of the Dipodidae family of the order Rodentia (Lioubimtseva & Henebry, 2009) that has physical characteristics similar to the Australian kangaroo (Macropus agilis). It is active only at night and uses hindlimbs to jump as it travels. The three-toed jerboa is widely distributed from the beaches of the Don River to the Caspian Sea in Russia, Turkmenistan, Uzbekistan, and northern Iran, and through Kazakhstan to the Irtysh River, Tuva, Mongolia, and northern China (IUCN, 2016). It has a relatively wider geographical distribution than most other species in the Dipodidae family (Yuan et al., 2018). The habitat of the three-toed jerboa is mostly high-altitude deserts and semi-deserts, but they can also be found in pine-covered sand dunes. In spring, the three-toed jerboa feeds on the vegetative parts of herbs and shrubs, as well as grass roots and bulbs. When the seeds start to ripen in the fall, this animal shifts its primary food source to seeds. Insects and larvae are also part of the daily diet of the three-toed jerboa. The three-toed jerboa is highly adapted to arid conditions and hardly drinks water in the wild (IUCN, 2016). The three-toed jerboa has an annual peak population in China from July to September, and the capture rate can reach more than 15% (Yuan et al., 2018; Ji et al., 2009). It not only harms desert steppe plant seeds, reducing the survival of wild plants and reducing vegetation coverage, but is also a potential cause of sandstorms. Due to its large activity range and wide distribution area, the three-toed jerboa is a risk factor for long-distance transmission of plague (Yersinia pestis), epidemic hemorrhagic fever, and other zoonotic diseases (Duan et al., 2010). This study found that the three-toed jerboa body surface parasites include a variety of fleas that can be naturally infected with Yersinia pestis, including: Xenopsylla skrjabini, Coptopsylla lamellifer ardua, Neopsylla identatiformis, Neopsylla pleskei orientalis, Leptopsylla pavlovskii, Mesopsylla hebes hebes, Frontopsylla wagneri, Ophthalmlmopsylla kiritschenkoi, Ophthalmlmopsylla praefecta praefecta and Citellophilus tesquorum sungaris (Yang et al., 2008). These fleas are also common parasitic fleas of other rodents, such as the squirrel (Sciuridae), chipmunk (Tamias sibiricus), ground squirrel (Spermophilus dauricus), red-cheeked ground squirrel (Spermophilus erythrogenys), striped-back hamster (Cricetulus barabensis), striped hairy-footed hamster (Phodopus sungorus), Brandt’s vole (Lasiopodomys brandtii), Eversmann’s hamster (Cricetulus eversmanni), desert hamster (Phodopus roborovskii), great gerbil (Rhombomys opimus), Mongolian gerbil (Meriones unguiculatus), midday gerbil (Meriones meridianus), Chinese white-bellied rat (Niviventer confucianus), Korean field mouse (Apodemus peninsulae), yellow steppe lemming (Eolagurus luteus), five-toed jerboa (Allactaga sibirica), Mongolian rabbit (Lepus tolai tolai), and domestic rodents like the Norway rat (Rattus norvegicus), house mice (Mus musculus), and their natural enemies such as yellow weasel (Mustela sibirica), musked polecat Mustela eversmanii), red fox (Vulpes vulpe), sand fox (Vulpes corsac), and domestic cat (Felis catus; Yang et al., 2008). In the wild, the habitats of rodents and their natural enemies overlap, and with infected fleas parasitizing all of these animals, the risk of long-distance transmission of diseases such as plague can be extremely high. In China, the three-toed jerboa is mainly distributed in desert and semi-desert ecosystems, which are relatively simple in structure, fragile in ecological function, and relatively sensitive to climate change (Yuan et al., 2018). In this study, the MaxEnt model, combined with ArcGIS graphics, was used to further clarify the impact of climate change on the distribution area of the three-toed jerboa. The past and present distribution range of the three-toed jerboa was simulated, the future distribution range was predicted under different climatic conditions (RCP4.5 and RCP8.5 scenarios), and then the main environmental factors and thresholds affecting the distribution of the three-toed jerboa were analyzed. This study provides insights into the risk of harmful rodents in an arid environment subject to “climate migration,” and can provide a scientific basis for controlling these rodents under future climate warming trends.

Materials and Methods

Source of data on the distribution of the three-toed jerboa

The distribution data of the three-toed jerboa was obtained from the Global Biodiversity Information Facility (http://www.gbif.org), the National Specimen Information Infrastructure (http://www.nsii.org.cn), a review of existing literature, and field data accumulated by our research group over the past 21 years. A total of 646 distribution points of Dipus sagitta were obtained. The buffer module of the ArcGIS software was used to eliminate the distributed data points, ensuring that there was only one distribution point in each grid to avoid the over-fitting phenomenon caused by too many distribution points. The spatial resolution of environmental data was 2.5 min (about 4.5 km), and the buffer diameter was set to 10 km. When the distance between two distribution points was less than 10 km, the buffer area overlapped, and only one distribution point was reserved (Wang, Xu & Li, 2017). After this process, a total of 216 effective distribution points were reserved (Fig. 1). The longitude and latitude coordinates of the sample were then converted into .csv format, which was used to construct the MaxEnt Model.

Figure 1 Geographic distribution of the three-toed jerboa in the present period.

Different colors represent different altitudes, and the black dots represent the actual distribution of three-toed jerboas.

Note: For the distribution data of Dipus sagitta, please refer to the supporting materials.

Environmental data sources

The 19 bioclimatic variables collected in this study (Table 1) were all derived from the World Climate Database (http://www.worldclim.org/). The coordinate system was WGS84, the grid size was 25 km2, and the data spatial resolution was 2.5 min. The time range of contemporary climate data was from 1960 to 2020. The Last Glacial Maximum (LGM, about 21,000 years ago), mid-Holocene (about 6,000 years ago), and future climate scenarios used the CCSM4 universal climate system model developed by the National Center for Atmospheric Research (NCAR). For future climate data, two greenhouse gas emission scenarios, RCP4.5 and RCP8.5 (RCP = Representative Concentration Pathways), from the Fifth Assessment Report of the IPCC were selected to represent a low and high impact of rising greenhouse gas concentrations on the future climate, respectively.

Table 1 Environment factor variables used in the MaxEent model.

Variable	Description	
Bio01	Mean annual temperature	
Bio02	Mean diurnal range	
Bio03	Isothermality	
Bio04	Temperature seasonality (standard deviation)	
Bio05	Max temperature of warmest month	
Bio06	Min temperature of coldest month	
Bio07	Temperature annual range (Bio05-Bio06)	
Bio08	Mean temperature of wettest quarter	
Bio09	Mean temperature of driest quarter	
Bio10	Mean temperature of warmest quarter	
Bio11	Mean temperature of coldest quarter	
Bio12	Annual precipitation	
Bio13	Precipitation of wettest month	
Bio14	Precipitation of driest month	
Bio15	Coefficient of variation of precipitation seasonality	
Bio16	Precipitation of wettest quarter	
Bio17	Precipitation of driest month	
Bio18	Precipitation of warmest quarter	
Bio19	Precipitation of coldest quarter	

Filtering and processing of environment variables

The LGM, mid-Holocene, current, and future climate scenarios all included 19 bioclimatic variables (Table 1). Since some of the bioclimatic variables are highly correlated, in order to avoid overfitting of the model caused by the multicollinearity of environmental variables (Lebedev et al., 2018; Michael, 2003), DIVA-GIS 7.5 software (http://www.diva-gis.org) was used to extract information on the 19 climate variables at 216 distribution points. R software was used to carry out a Pearson correlation analysis (Pearson et al., 2006). Environmental variables with correlation coefficient r < 0.8 were retained, and environmental variables with correlation coefficient r > 0.8 were selected to have a larger contribution rate in the initial model test (Fig. 2). After final screening, seven environmental variables were obtained: mean annual temperature (Bio01), mean diurnal range (Bio02), temperature seasonality (Bio04), mean temperature of wettest quarter (Bio08), annual precipitation (Bio12), coefficient of variation of precipitation seasonality (Bio15), and precipitation of coldest quarter (Bio19).

Figure 2 Pearson correlation analysis of 19 bioclimatic variables.

Different colors represent different correlation coefficients.

Model building

Data from the 216 Dipus sagitta distribution points and seven climate variables in different periods were introduced into MaxEnt 3.3.3 to predict the potential distribution of Dipus sagitta under climate change conditions. A total of 75% of the distribution points of Dipus sagitta were randomly selected as training sets for model construction, and the remaining 25% were used as test sets for model verification. Cross-validation was adopted, meaning the species data were randomly divided into 10 parts, with one part randomly selected as the test set each time, and the remaining nine parts used as training sets, repeated ten times. By default, the maximum number of iterations was 500 and the maximum number of background points was 10,000.

A key parameter setting in MaxEnt is feature class, which has five feature types: linear feature (L), quadratic feature (Q), fragmented feature (H), product feature (P), and threshold feature (T). With more distribution points, MaxEnt uses more of these feature types by default (Elith, Kearney & Phillips, 2010; Wang et al., 2017; Kong, Li & Zou, 2019), with little impact on overall performance. Because the distribution points of the selected species in this study were >80, the default feature selection of MaxEnt was selected.

This data was used to plot environmental variable response curves to evaluate the impact of each climate variable on the model prediction results. Receiver operating characteristic (ROC) curves were created and the jack-knife method was used to assess the importance, or contribution, of each environmental variable to distribution gain. The jack-knife method calculates the training scores when simulating “with only one variable,” “without variables,” and “with all variables,” respectively. When the “with only one variable” had a higher score, it indicated that the environmental factor being used as the variable had a higher predictive ability and thus a greater contribution to species distribution. When the training scoring ability of the “with only one variable” model decreased, it indicated that the variable had more unique information and was more important to species distribution (Phillips & Miroslav, 2008). Using this method, the dominant factors affecting the species distribution were determined, and the remaining parameters were kept at their default settings. The model then output the results in the form of logistic output, from which the ASC II (American Standard Code for Information Interchange) file of the average of 10 operations were selected. Raster values giving the distribution probabilities of the species were used as logical values reflecting the extent to which each grid in the target area met the actual ecological niche of the species. These results were then input into ArcGIS 10.2 software (Yang et al., 2013). The automatic classification method in the reclassification tool was used to divide the suitability of the distribution areas into four grades: non-suitable area, somewhat suitable area, moderately suitable area, and highly suitable area.

Geospatial analysis

ArcGIS 10.2 was used to count the different grades of suitable areas in different periods, and SDM toolbox 2.4 tool was used to calculate the potential distribution areas of Dipus sagitta in different periods (Brown, 2014). Using the “Reclass” function in ArcGIS 10.2, the grid values of suitable (including somewhat, moderate, and highly suitable areas) and non-suitable areas of Dipus sagitta were modified to 1 and 0 respectively, then SDM toolbox was added, and the subdirectory of “MaxEnt Tools” in the “SDM Tools” module was selected. The “Distribution Changes Between Binary SDMs” tool was used to calculate changes in potential distribution areas in each period (LGM maximum to mid-Holocene, mid-Holocene to current, current to RCP 4.5-2070, and current to RCP 8.5-2070) and obtain the distribution expansion area, stable area, and contraction area.

Model accuracy evaluation

Receiver operating characteristics (ROC) take every value of the prediction result as a possible judgment threshold and calculates the corresponding sensitivity and specificity, and then evaluates the accuracy of the model. Because the area under the ROC curve (AUC) is not affected by the judgment threshold, it is recognized as the best index to evaluate the prediction accuracy of the model. The value range of AUC is [0, 1]. The larger the value, the farther it is away from the random distribution, indicating a more accurate prediction effect. The evaluation criteria are: 0.7–0.8 is more accurate, 0.8–0.9 is accurate, and 0.9–1.0 is very accurate (Phillips, Anderson & Schapire, 2006).

Because the subject working curve generated by MaxEnt was fuzzy, to improve the clarity, the origin picture numerical drawing toolbox was used to extract the result of the MaxEnt subject working curve. This extracted result was then imported into MATLAB and the plot function was used to redraw the ROC curve. This study also chose a partial receiver operating character (PROC) measurement method to evaluate the prediction performance of the model (http://shiny.conabio.gob.mx:3838/nichetoolb2/). Finally, the model was constructed based on the distribution data of all species. The PROC method uses AUC ratio to evaluate the model. An AUC ratio >1 indicates that the model has better relative random prediction results, while AUC ratio ≤ 1 indicates that the model has worse relative random prediction results (Fan et al., 2019). PAUC was calculated by NicheToolbox (http://shiny.conabio.gob.mx:3838/nichetoolb2/) with 1,000 iterations, E = 0.05.

Laboratory animal ethics

The animal study was reviewed and approved by the Research Ethics Review Committee of Inner Mongolian Agricultural University (NND2023081).

Results and analysis

MaxEnt model accuracy verification

The MaxEnt model was used to simulate the potential habitat of Dipus sagitta, and the ROC was used as a measure of the prediction accuracy of the model. The evaluation results of ROC show that the average AUC (referring to the area surrounded by ROC and abscissa axis) of the prediction model of suitable living area of Dipus sagitta was greater than 0.9 (Fig. 3), the average AUC value of the evaluation results of the model PROC performance test was 0.906 (P < 0.001) (Fig. 4), and the AUC ratio was 1.813, which is much greater than 1, indicating that the evaluation results of the prediction model and random model were significantly different, and the performance of the prediction model was significantly stronger than the random model. This suggests that the mean value of the MaxEnt model can be used to reflect the distribution of species with high accuracy.

Figure 3 AUC results of the MaxEnt model under current conditions.

The red line in the plot represents the average AUC value, and the blue area represents the standard deviation.

Figure 4 pROC test of MaxEnt model for three-toed Jerboa.

The shaded bars with bell-shaped curve indicate the frequen- cy distribution of the ratios between AUC from model prediction and AUC random, while the bell-shaped curve on the left represents the AUC ratios for random models.

The importance of environmental variables affecting the distribution of the three-toed jerboa

The influence of seven environmental variables on the potential future distribution area of the three-toed jerboa was assessed using the jack-knife method (Table 2). The top three variables by percentage contribution were: precipitation of the coldest quarter (Bio19, 38.77%), for which the most suitable range was 0–22.79 mm; temperature seasonality (standard deviation; Bio04, 26.15%), for which the most suitable range was 86.72–250; and mean annual temperature (Bio01, 23.76%), for which the most suitable range was 1.37–19.73 °C. The cumulative contribution rate of these three factors was 88.68%. The top three permutation importance values were: the standard deviation of temperature (Bio04, 32.01%), mean annual temperature (Bio01, 30.21%), and precipitation of the coldest quarter (Bio19, 19.69%). The cumulative importance value of these three factors was 81.92%.

Table 2 The importance of seven environmental variables to the distribution of three-toed jerboa.

	Environment variable	PC/%	PI/%	TRGO	TRGW	TGO	TGW	AUCGO	AUCGW	
Bio19	Precipitation of coldest quarter	38.77	19.69	1.089	2.478	0.7840	2.232	0.8201	0.9604	
Bio04	Temperature seasonality	26.15	32.01	0.8908	2.451	0.9542	2.080	0.8437	0.9530	
Bio01	Mean annual temperature	23.76	30.21	1.157	2.457	1.075	2.173	0.8708	0.9533	
Bio02	Mean diurnal range	7.470	3.520	0.6770	2.531	0.4513	2.206	0.7610	0.9592	
Bio12	Annual precipitation	2.920	12.73	0.8306	2.511	0.7996	2.147	0.8336	0.9580	
Bio15	Coefficient of variation of Precipitation seasonality	0.5201	0.2401	0.5851	2.541	0.3317	2.223	0.7072	0.9601	
Bio08	Mean temperature of wettest quarter	0.4401	1.570	0.7405	2.533	0.6743	2.201	0.8095	0.9591	
Notes.

PC Percentage contribution

PI Permutation importance

TRGO Training gain with only

TRGW Training gain without

TGO Test gain with only

TGW Test gain without

AUCGO with only

AUCGW without

The simulation results of the MaxEnt model showed that, among the seven environmental variables in this study, both precipitation and temperature had a certain degree of influence on the distribution area of the three-toed jerboa. The ranking of contribution rates showed that the precipitation factor was more important, but replacement importance value results showed that the temperature factor was more important. These results suggest that the two main factors affecting the contemporary geographic distribution of the three-toed jerboa are precipitation and temperature. Precipitation of the coldest quarter was the environmental factor with the highest contribution rate. The simulation results showed that the most favorable range of precipitation of the coldest quarter for the survival of the three-towed jerboa was 0–22.79 mm, which is in line with this animal being highly adapted to arid conditions (IUCN, 2016).

Potential suitable habitats of the three-toed jerboa in different climatic backgrounds

The potential distribution area of the three-toed jerboa under the current climate background was simulated by the MaxEnt model (Fig. 5). This area covered almost all the sample points, indicating that the simulated distribution area of the three-toed jerboa was in close agreement with its actual distribution. The distribution of highly suitable areas was relatively concentrated (brown area), with a total area of 3.25 ×106 km2. This region mainly included China’s Xinjiang region, central and western Inner Mongolia, Shaanxi, Shanxi, and Ningxia. In Mongolia, the highly suitable areas were mainly distributed in the western region. There were also sporadic distributions in Kyrgyzstan. The moderately suitable areas (dark green area) were mainly concentrated in southern Kazakhstan, western Uzbekistan, eastern Turkmenistan, and along the edge of the highly suitable areas, with a total moderately suitable area of 3.51 ×106 km2. The somewhat suitable areas (yellow area) were scattered throughout northern Kazakhstan, southern Turkmenistan, southern Uzbekistan, and the borders between Russia, China, and Mongolia. The simulated distribution area also showed areas suitable for the survival of the three-toed jerboa in the Midwest region of North America, but there is no actual distribution of the three-toed jerboa in that region. The main species of rat in the Midwest region of North America is the kangaroo rat (Dipodomys spectabilis), which is very similar to the three-toed jerboa in both morphological and physiological characteristics (Li et al., 2007).

Figure 5 Potential distribution of the three-toed jerboa in the current climate.

This figure represents the suitable distribution area of the three-toed jerboa in the current climate background, where white represents the non-suitable area, yellow represents the somewhat suitable area, dack green represents the moderately suitable area, and brown represents the highly suitable area.

The suitable distribution area of the three-toed jerboa varied greatly across the Last Glacial Maximum, the mid-Holocene, current climate conditions, and future climate scenarios up to the 2070s (Table 3). The highly suitable area for the three-toed jerboa in the Last Glacial Maximum was the smallest, at 1.72 ×106 km2, and then it continued to spread and increase, reaching 3.25 ×106 km2 in the current climate period, an increase of 88.95%. By the 2070s, under the RCP8.5 scenario, highly suitable areas could be reduced to 2.6 ×106 km2. During the Last Glacial Maximum, the moderately suitable area of the three-toed jerboa was small, 2.03 ×106 km2, after which it began to increase. The moderately suitable area in the mid-Holocene expanded to 4.17 ×106 km2, an increase of 105.49%. From the mid-Holocene to the 2070s under RCP8.5, the area of suitable habitat for the three-toed jerboa shrank slightly, reducing by 0.42 ×106 km2 to 3.75 ×106 km2, which was still higher than in the Last Glacial Maximum. The somewhat suitable area for the three-toed jerboa in the mid-Holocene was the largest at 9.3 ×106 km2, an increase of 99.14%, or 4.63 ×106 km2 compared with the Last Glacial Maximum. After this, the somewhat suitable area for three-toed jerboa shrank gradually, and could reduce to 7.26 ×106 km2 in the 2070s under RCP8.5.

Table 3 Suitable distribution area of different grades of the three-toed jerboa under different climatic background (×106km2).

Period	Climate scenario	Suitable distribution area of different grades	
		Unsuitable area	Low suitable area	Moderate suitable area	High suitable area	
LGM	–	140.45	4.67	2.03	1.72	
Middle Holocene	–	132.51	9.3	4.17	2.9	
current	–	134.18	7.95	3.51	3.25	
2070s	RCP4.5	134.61	8.24	3.38	2.65	
2070s	RCP8.5	135.27	7.26	3.75	2.60	

Spatial changes in the total suitable habitat of the three-toed jerboa under different climate scenarios

To better analyze changes in the suitable area for the three-toed jerboa, the three gradients of suitable area (somewhat, moderately, and highly) were considered together as the total suitable area. The current three-toed jerboa distribution layer was superimposed on the Last Glacial Maximum and the mid-Holocene prediction layer, and the differences in distribution were analyzed (Table 4, Figs. 6, 7). The results showed that from the Last Glacial Maximum to the mid-Holocene, the spatial pattern change rate of the three-toed jerboa was 93.91%, and the expansion rate reached 111.67%. The total suitable habitat has increased on a large scale. Kazakhstan, China, Turkmenistan, Afghanistan, Iran, and Mongolia all experienced varying degrees of expansion. The change rate of the spatial distribution pattern of the three-toed jerboa from the mid-Holocene to the present was 10.19%, with a shrinkage rate of 19.11%. The shrinking areas were mainly concentrated in Afghanistan and Iran, and there were also sporadic shrinkages in central and western China. The expansion rate was 5.49%, concentrated primarily at the junction of Afghanistan, Iran, and Turkmenistan, with some expansion in the southern area of Afghanistan, indicating that the three-toed jerboa began to migrate southward.

Table 4 Changes of suitable distribution areas of Three-toed jerboa in different climatic contexts.

period	Climate scenario	area/(×106 km2)	(%)	
			expansion	Stable	construction	change	expansion rate	Stable rate	construction rate	change rate	
LGM		–									
mid- Holocene	–	9.43	6.94	−1.5	7.93	111.67	82.19	−17.76	93.91	
current	–	0.9	13.25	−3.13	−1.67	5.49	80.87	−19.11	−10.19	
current		–									
2070s	RCP4.5	0.85	13.8	−1.61	−0.43	5.77	93.81	−10.94	−2.92	
2070s	RCP8.5	0.99	12.85	−1.86	−1.09	6.73	87.35	−12.64	−7.41	

Figure 6 Changes in the distribution pattern of suitable three-toed jerboa areas under climate change scenarios from the Last Glacial Maximum to the mid-Holocene.

This figure shows changes in the suitable distribution area of the three-toed jerboa from the Last Glacial Period to the middle of the Holocene period. Light blue represents the unchanged area, dark blue represents the expansion area, and red represents the contraction area.

Figure 7 Changes in the distribution pattern of suitable three-toed jerboa areas under climate change scenarios from the mid-Holocene to the current period.

This figure shows changes in the suitable distribution area of the three-toed jerboa from the mid-Holocene to the current period. Light blue represents the unchanged area, dark blue represents the expansion area, and red represents the contraction area.

The current three-toed jerboa distribution layer was superimposed on the forecast layer under the two carbon emission scenarios for the 2070s, RCP4.5 and RCP8.5, and the change trend was analyzed (Table 4, Figs. 8, 9). Under these future climate change scenarios, the suitable habitat of the three-toed jerboa generally shrinks in the east, west, and south and expands to the north. The areas of expansion are mainly located at the junction of Mongolia and Russia, the junction of eastern Inner Mongolia and Russia, and the northern border of Kazakhstan, while the shrinking distribution areas are mainly concentrated in the southern border of Inner Mongolia, China, eastern Jilin Province, western Kazakhstan, Uzbekistan, Kyrgyzstan, and southern Turkmenistan. Under the RCP4.5 emission scenario, the shrinkage rate is 1.61%, and under the RCP8.5 emission scenario, the shrinkage rate is 1.86%. These results indicate that in order to adapt to high-concentration CO2 emissions, in future climate warming scenarios, the distribution area of the three-toed jerboa will migrate and expand to higher latitudes.

Figure 8 Changes in the distribution pattern of suitable three-toed jerboa areas under climate change scenarios of the current period to the 2070s under RCP4.5.

This figure shows changes in the suitable distribution area of the three-toed jerboa from the current period to the RCP4.5 climate scenario in 2070. Light blue represents the unchanged area, dark blue represents the expansion area, and red represents the contraction area.

Figure 9 Changes in the distribution pattern of suitable three-toed jerboa areas under climate change scenarios of the current period to the 2070s under RCP8.5.

This figure shows changes in the suitable distribution area ofthe three-toed jerboa from the current period to the RCP8.5 climate scenario in 2070. Light blue represents the unchanged area, dark blue represents the expansion area, and red represents the contraction area.

Discussion

In this study, the MaxEnt model was used to predict changes in the suitable habitat of the three-toed jerboa in different climatic periods. There may have been ice sheets on the Qinghai-Tibet Plateau during the period investigated, which would have greatly affected the survival of three-toed jerboa (Jiang, Wang & Lang, 2002). In addition, the Last Glacial Maximum was a period of colder temperature and lower humidity, with an average temperature 5.3 °C lower than the present, and land precipitation totaling 71% of the current level (Yang, 1990). Therefore, the suitable habitats for most species were greatly reduced during this period. As a drought-tolerant species, the southward migration of the three-toed jerboa during this period was also a manifestation of its response to these climate changes. In the mid-Holocene, the suitable area for the three-toed jerboa expanded greatly, and the distribution pattern began to be similar to the contemporary distribution area. This expansion may be related to two factors. First, a large amount of geological data show that the climate in the mid-Holocene was similar to the contemporary climate. The ice sheet in the north melted, the soil thawed, and the temperature increased. Therefore, the distribution area of the three-toed jerboa expanded to the south and north. Second, the mid-Holocene is a geological historical period most closely related to modern humans, and the utilization of land, forests, grasslands, and water resources by humans gradually became an important factor affecting environmental changes (Li et al., 2007). Human activity has exacerbated the occurrence of desertification (Yang et al., 2009), leading to an increase in the suitable habitat for the three-toed jerboa.

Future global warming caused by an increase in atmospheric CO2 should be similar to the warming effect seen in the mid-Holocene (Zheng et al., 2004). The map of the change in suitable areas in each time period shows that the expansion area from the mid-Holocene to the modern period is basically the same as the contraction area from the current to the future period. The area of contraction from the mid-Holocene to the modern period is also very close to the expansion area from the current period to the future period. This map also shows that the climatic environment in the future is likely to be very similar to the mid-Holocene, and the impact on the three-toed jerboa will likely be similar. The distribution area of the three-toed jerboa, especially highly suitable areas, show significant reductions in the current period, with greater reductions seen in future climate scenarios. This suggests that climate warming will have a significant impact on the distribution of the three-toed jerboa.

Species are distributed in areas with the most suitable climatic conditions, so as the climate changes, the distribution area of animals will respond accordingly (Kausrud et al., 2010). The results of this study indicate the suitable distribution area for the three-toed jerboa will spread to the north under future climatic conditions and that the main environmental factor affecting the distribution of the three-toed jerboa is precipitation in the coldest season, or the total amount of snowfall. Snow is also known to impact the dynamics of plague transmission (Zhang et al., 2003). Winter snowfall directly affects soil moisture in the spring, which affects vegetation growth and the survival and reproduction of media organisms. This cascading effect of precipitation on plague epidemics can be explained by the trophic cascade hypothesis. The trophic cascade hypothesis is the basic theory demonstrating the relationship between plague epidemics and climatic factors. It postulates that increased precipitation enables lush plant growth, provides sufficient food for rodents, increases rodent population density and abundance, and thus promotes plague epidemics (Davis et al., 2007).

In recent years, plague research has mainly focused on the main plague host and neglected the role of secondary hosts in plague epidemics. The three-toed jerboa is a relatively common infected animal in plague foci, and many infected individuals have been identified in China (Huang et al., 2010; He et al., 2011). Under future climate change scenarios, the potential suitable three-toed jerboa habitat will expand to the north. The future increase in ambient temperature will also be conducive to the reproduction of fleas, which will likely increase the flea index and increase the probability of plague transmission. There is a potential risk of long-distance transmission of plague in the northern expansion of the three-toed jerboa habitat. Climate warming will lead to the migration of wild animals, which could promote cross-species transmission of pathogens such as Yersinia pestis and increase the risk of pathogen transmission and disease outbreaks. Monitoring pathogen transmission among animals in the newly-added suitable distribution area of the three-toed jerboa should be further strengthened to prevent the outbreak and spread of epidemic diseases.

Supplemental Information

Supplemental Information 1 Distribution

Thanks to Andreas Wilkes for touching up the article.

Additional Information and Declarations

Competing Interests

Author Contributions

Animal Ethics

Data Availability

The authors declare there are no competing interests.

Fan Bu conceived and designed the experiments, analyzed the data, prepared figures and/or tables, authored or reviewed drafts of the article, and approved the final draft.

Xiuxian Yue conceived and designed the experiments, analyzed the data, prepared figures and/or tables, authored or reviewed drafts of the article, and approved the final draft.

Shanshan Sun conceived and designed the experiments, prepared figures and/or tables, authored or reviewed drafts of the article, and approved the final draft.

Yongling Jin conceived and designed the experiments, prepared figures and/or tables, authored or reviewed drafts of the article, and approved the final draft.

Linlin Li conceived and designed the experiments, analyzed the data, authored or reviewed drafts of the article, and approved the final draft.

Xin Li performed the experiments, authored or reviewed drafts of the article, and approved the final draft.

Rong Zhang performed the experiments, analyzed the data, prepared figures and/or tables, authored or reviewed drafts of the article, and approved the final draft.

Zhenghaoni Shang performed the experiments, prepared figures and/or tables, authored or reviewed drafts of the article, and approved the final draft.

Haiwen Yan performed the experiments, authored or reviewed drafts of the article, and approved the final draft.

Haoting Zhang performed the experiments, authored or reviewed drafts of the article, and approved the final draft.

Shuai Yuan performed the experiments, authored or reviewed drafts of the article, and approved the final draft.

Xiaodong Wu performed the experiments, authored or reviewed drafts of the article, and approved the final draft.

Heping Fu performed the experiments, authored or reviewed drafts of the article, and approved the final draft.

The following information was supplied relating to ethical approvals (i.e., approving body and any reference numbers):

The animal study was reviewed and approved by the Research Ethics Review Committee of Inner Mongolian Agricultural University (Approval Codes: NND2023081; Approval Date: 6 March 2023).

The following information was supplied regarding data availability:

The raw data are available as a Supplemental File.

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
