# Peer review of "Would future climate warming cause zoonotic diseases to spread over long distances?"

_PeerJ, doi:10.7717/peerj.16811_

## Round 0.1 · original submission · Major Revisions

Dear esteemed authors,

I would like to begin by acknowledging the potential significance of your manuscript for the species. After receiving two rounds of revisions, both reviewers agree that it is a compelling piece of research. However, they also agree that Mayor Revisions are necessary before it can be accepted for publication.

To improve the manuscript, it is essential to provide a more detailed account of data collection and cleaning processes, model parameters, and variable selection. Additionally, further analysis is required, and the writing needs to be refined. I also recommend that you have a fluent speaker review your English to ensure it is polished and clear.

I appreciate your commitment and hard work on this manuscript and look forward to seeing the revised version.

Best regards,

Armando Sunny

·

Basic reporting

The manuscript presents original research that is relevant to the journal's scope. It examines the potential impact of climate change on the distribution of the three-toed jerboa using species distribution modeling. However, the language and writing style needs significant improvements for coherence, conciseness, and clarity before the manuscript would be suitable for publication.
The introduction provides context on climate change impacts and species distribution modeling. However, the literature review is limited and several key statements lack supporting references. The methods follow the overall structure expected but sections need to be reorganized and streamlined for clarity. Figure quality and resolution should be improved.

Experimental design

The research aims to address a meaningful question about how climate change may affect a species’ habitat. However, the specific research questions and knowledge gaps being addressed require clarification.
The methods are described but lack sufficient detail for replication. More information is needed on data collection, cleaning, model parameters, and selection of variables. The statistical methods appear sound but details are lacking to fully assess their validity. Reporting additional evaluation metrics beyond AUC would strengthen confidence in the model's robustness.

Validity of the findings

The results suggest climate change may significantly impact habitat suitability for the study species. However, the data underlying these findings have not been provided, limiting the ability to evaluate their strength. The conclusions are linked to the research question but are overly broad given the limitations of the data and analysis. Meaningful replication of the study is needed before the conclusions can be considered robust.

Additional comments

Overall, this manuscript addresses an important topic and has the potential to make a useful contribution to the literature. However, major improvements are still needed to adequately describe the methods, analyze the data, present the results, discuss the implications, and support the conclusions. Providing the data and clarifying the analysis would allow for a more thorough evaluation of validity before publication in a reputable journal. With significant revisions that address the identified limitations, this manuscript could eventually make a meaningful contribution as part of a well-developed study.

Reviewer 2 ·

Basic reporting

poor English used principally in intrtoduction and discussion.

could improve the literature, if you add more references of examples on SDM in small mammals

improve the resolution of figures (2 and 3)

good article structure

relevant results to hypotheses.

delete table 1

rewrite discussion section and introduction (lines 76 to 95; This should include the importance of previous studies and how the impact has been in the different climate scenarios if it increases or decreases, if they migrate or not)

review the way to cite an author (s) throughout the text there are many discrepancies. I suggest revising the editorial guidelines of the journal.

Explain why it is losing its habitat, what are the causes for which this species is in danger? Similarly, explain why they conducted this study (justification? lines 99-108)

in references, check that the journal names are homogenized (e.g., line 361-362 change "change" by "Change"; line 382 change "bmc" by "BMC")

line 389 Complete the reference

Experimental design

In methods, it is necessary to carry out a collinearity analysis with tools such as VIF and kuenm in R; In addition, for the evaluation of the model, apply a pROC test and integrate another GCM

Validity of the findings

no comment

---

## Round 0.2 · Minor Revisions

Dear Authors,

Following a thorough evaluation, it has been noted that some minor corrections are required before your manuscript can be finalized for acceptance in PeerJ.

Your cooperation in addressing these adjustments is highly appreciated.

Sincerely,

Armando Sunny

·

Basic reporting

The paper investigates an important research question related to the potential impact of climate change on the spread of zoonotic diseases by studying the change in suitable habitat distribution of the three-toed jerboa, which carries disease-spreading fleas. the authors have adequately addressed the requested revisions. I appreciate the authors' efforts and the paper is now suitable for publication. I should only mention the following points that could be improved:
The abstract does not provide enough quantitative details on the degree of distribution range changes predicted and does not provide the specific climate variables examined or the key findings of which variables were most impactful. Listing the main variables driving distribution shifts would strengthen the Abstract.
Authors should add the value of AUC in the “abstract” which reflects the model performance.

Experimental design

The methods and investigation are rigorous, utilizing an established species distribution modeling approach in MaxEnt. Parameters and implementation are described sufficiently to enable replication. Data processing and analyses also seem sound.

Validity of the findings

The data overall seem robust - derived from various reputable sources. The statistical analyses are clearly explained and seem appropriate. MaxEnt modeling results also show strong predictive performance based on AUC and other accuracy measures.

Reviewer 2 ·

Basic reporting

NO COMMENT

Experimental design

NO COMMENT

Validity of the findings

NO COMMENT

---

## Round 0.3 · accepted · Accept

Dear Authors,

I am pleased to inform you that reviewers have concurred on the appropriateness of the corrections implemented in your manuscript. As a result, it is with great satisfaction that I announce the acceptance of your work for publication in PeerJ. I extend my heartfelt gratitude for the meticulous attention to detail demonstrated in your revisions.

Thank you for selecting PeerJ as the platform for sharing your compelling and valuable research. Your contribution is integral to advancing our understanding and promoting the conservation of this species.

Warm regards,

Armando Sunny